# Schema-adaptable Knowledge Graph Construction

**Hongbin Ye**[1,2,4], **Honghao Gui**[1,4], **Xin Xu**[1,4], **Xi Chen**[3], **Huajun Chen**[1,4], **Ningyu Zhang**[1,4] [*]

[1] Zhejiang University [2] Zhejiang Lab [3] Platform and Content Group, Tencent
[4] Zhejiang University - Ant Group Joint Laboratory of Knowledge Graph
yehongbin@zhejianglab.com, jasonxchen@tencent.com,
{guihonghao,xxucs,huajunsir,zhangningyu}@zju.edu.cn

## Abstract

Conventional Knowledge Graph Construction (KGC) approaches typically follow the static information extraction paradigm with a closed set of pre-defined *schema*. As a result, such approaches fall short when applied to dynamic scenarios or domains, whereas a new type of knowledge emerges. This necessitates a system that can handle evolving schema automatically to extract information for KGC. To address this need, we propose a new task called schema-adaptable KGC, which aims to continually extract entity, relation, and event based on a dynamically changing schema graph without re-training. We first split and convert existing datasets based on three principles to build a benchmark, i.e., horizontal schema expansion, vertical schema expansion, and hybrid schema expansion; then investigate the schema-adaptable performance of several well-known approaches such as Text2Event, TANL, UIE and GPT-3.5. We further propose a simple yet effective baseline dubbed ADAKGC, which contains schema-enriched prefix instructor and schema-conditioned dynamic decoding to better handle evolving schema. Comprehensive experimental results illustrate that ADAKGC can outperform baselines but still have room for improvement. We hope the proposed work can deliver benefits to the community[1].

## 1 Introduction

Knowledge Graph Construction (KGC), typically through information extraction, has enjoyed widespread empirical success and can provide back-end support for various NLP tasks, such as question answering (Saxena et al., 2020; Shang et al., 2022; Zhang et al., 2022a), commonsense reasoning (Yasunaga et al., 2021; Zhang et al., 2022d) etc. Traditional KGC tasks, including named entity recognition (NER) (Liu et al., 2021; Wang et al., 2021),

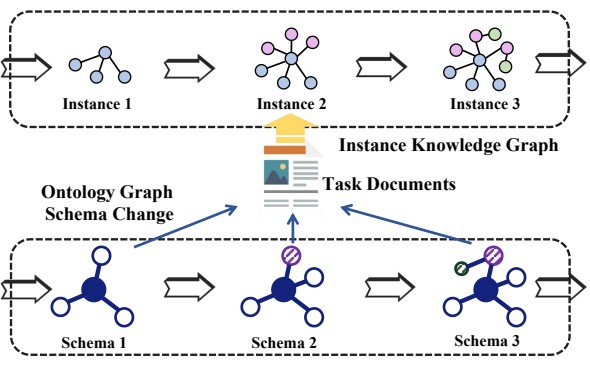

Figure 1: Knowledge Graph Construction (KGC) with dynamic updates of *schema*.

relation extraction (RE) (Chen et al., 2022a; Zheng et al., 2021; Yao et al., 2022) and event extraction (EE) (Huang et al., 2018; Liu et al., 2020; Lu et al., 2021; Lou et al., 2023) are "reactive", relying on static pre-defined *schema* from end-users. However, as shown in Figure 1, the schema may evolve along with scenario adaptation, making previous models challenging to utilize without re-training.

Note that existing information extraction systems can only handle a fixed number of classes by pre-defined schema and performing once-and-for-all training on a fixed framework. It is desirable to respond to changes (e.g., evolving schema) to existing KGs, making the system act more "proactively" like humans who can handle flexible knowledge updates. Early, several approaches introduce incremental learning (Cao et al., 2020; Wang et al., 2019; Shen et al., 2020; Cui et al., 2021b) to learn new classes continually. In this case, the extraction system learns from the class incremental data stream but usually suffers significant performance degradation on the old class when adapting to the new class. Stated differently, previous studies put emphasis on struggling against catastrophic forgetting (Thrun, 1998). However, for the schema-evolving scenarios, the dynamic generalizability of extraction models plays a vital role and needs to be

---

[*] Corresponding author.

[1] Code and datasets available at https://github.com/zjunlp/AdaKGC.

inspected from an ontology evolution perspective.

Therefore, we propose a novel KGC task dubbed schema-adaptable KGC, where the models are required to have the ability to represent and adapt to complement knowledge extraction. We first construct datasets according to three principles of evolutionary schema directions (*Horizontal Schema Expansion*, *Vertical Schema Expansion*, and *Hybrid Schema Expansion*) on three tasks of NER, RE[2], and EE. Through empirical analysis, we notice that approaches of Text2Event (Lu et al., 2021), TANL (Paolini et al., 2021), UIE (Lu et al., 2022), and GPT-3.5(Ouyang et al., 2022) cannot effectively extract the information given complex evolving schema. We argue that the major issues lie in the following: 1) How to learn dynamic and generalizable schema representations as conditions for extraction; 2) How to precisely extract new instances constrained with newly updated schema.

To this end, we propose a simple baseline dubbed ADAKGC, which introduces *Schema-enriched Prefix Instructors (SPI)* to represent and transfer the learnable schema-specific knowledge. At each iteration stage, we linearly convert from the current schema graph to learnable prompts, initialized with the ontology name and connected to task-specific prefixes. To encourage the decoder to understand the dynamic schema, we utilize a *Schema-conditioned Dynamic Decoding (SDD)* strategy that constructs a decoding path of schema-specific vocabulary to the output space. When the schema changes, we dynamically construct a new trie-tree to adjust the output space. Note that ADAKGC is model agnostic and can handle a variety of challenging schema evolution scenarios. We summarize the contribution of this work as:

- We introduce a new task of the schema-adaptable KGC to meet the schema evolution requirements, which is a new branch that has not been well-explored to the best of our knowledge.

- We propose a new baseline ADAKGC, which includes schema-enriched prefix instructors and schema-conditioned dynamic decoding strategy, and experimentally demonstrate the adaptability.

- We release the schema-adaptable KGC benchmark, which imposes new challenges and

---

[2]We regard RE as relational triple extraction in this paper.

presents new research opportunities for the NLP community.

## 2 Problem Statement and Overview

### 2.1 Background of KGC

KGC has been a promising research challenge (Lu et al., 2022; Zhang et al., 2022b), and existing benchmarks utilize a well-defined schema for directing knowledge graph construction, focusing on generating domain-specific knowledge graphs or aggregating heterogeneous structured databases. For example, FEW-NERD (Ding et al., 2021) consists of coarse-grained and fine-grained entity type definitions to locate and classify named entities from unstructured natural language. NYT (Riedel et al., 2010a) extracts relational triple instances specifically from textual data sources according to a specific taxonomy structure. ACE2005 (Ntroduction) identifies triggers and event types based on context, and each has its own event arguments, described in a slot-filling way. In addition, TAC-KBP (Ellis et al., 2014) is designed to leverage existing generic domain structured data sources and extend entity links employing descriptive text as additional information. OAEI (Euzenat et al., 2011) creates an integrated ontology based on an alignment between two or more existing ontologies or knowledge graphs. In this paper, we focus on the work of extracting knowledge instances from unstructured text, which is regarded as the schema-constraint prediction (structure prediction) task.

### 2.2 Definition of Schema-adaptable KGC

In the real world, the KGC system extracts structured knowledge from unstructured text and normalizes it to the instance graph according to a frequently adjusted schema. Given a set of schema graphs $\mathcal{S} = \{s_1, s_2, ..., s_n\}$, the task of schema-adaptable KGC is to generate a set of schema-constraint instances $\mathcal{G} = \{g_1, g_2, ..., g_n\}$ for each iteration. Suppose there is a model $\mathcal{M}_\theta = LM(\mathcal{D}_{train}^1 | S_1)$ trained on the initial training set, after which labeled data for updated schema are not available. A schema-adaptable data stream $\{\mathcal{D}^{(1)}, \mathcal{D}^{(2)}, ..., \mathcal{D}^{(N)}\}$ is provided to evaluate the adaptability of model for the dynamic updates of schema. Each $\mathcal{D}^{(k)}$ contains dev/test data $(\mathcal{D}_{dev}^k, \mathcal{D}_{test}^k)$ and schema graph $s_k$. Note that the model **will not be re-trained** but hope to pick up on the ability of information extraction with evolving schema. The challenge is that the model is

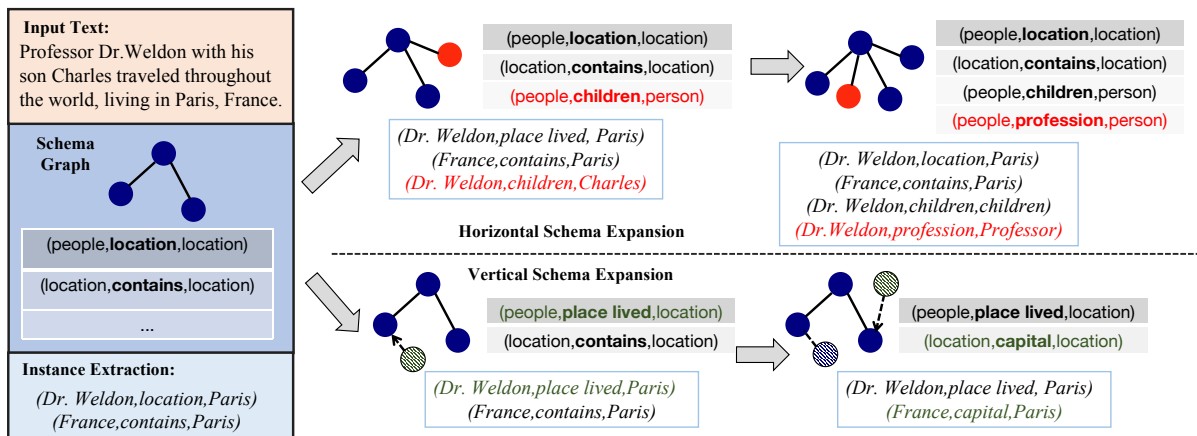

Figure 2: Illustration of schema-adaptable KGC with iterative schema evolution. Schema graph nodes (colored ●) are evolved by adding a new class node (colored ●) in horizontal schema expansion while is inherited by a new subclass node (colored ●) in vertical schema expansion.

expected to perform well in each iteration of the test set $\mathcal{D}_{\text{test}}^k$, which contains the golden instances changed for the updated schema.

## 2.3 Dataset Construction Process

As shown in Figure 2, we design three principles regarding different types of schema evolution and apply Algorithm 1 to build the dataset for evaluation: (1) **Horizontal Schema Expansion** requires the schema to add new class nodes of the same level, which can be considered a form of class-incremental learning without new classe instances as training data. Based on the generalization effect on the neighboring new classes, we can assess the transfer capabilities of the schema feature. (2) **Vertical Schema Expansion** requires the schema to add subclasses of father classes. Based on the generalization effect on subclasses, we can assess the inheritance and derivation capabilities of the schema feature. (3) **Hybrid Schema Expansion** requires the schema to randomly expands nodes horizontally or vertically at each iteration, which summarizes schema graphs and represents their potential co-evolutionary pattern. More details are in Appendix A.1, besides the above structural extensions, we further explore analogous node replacement from the perspective of semantics.

## 2.4 Schema-adaptable KGC Benchmark

There are two challenges for schema-adaptable KGC. Firstly, since the schema is updated in each iteration, the schema evolution information needs to be dynamically injected into the model. Secondly, since the output target of KGC is often demand-specific, the extraction results should be adaptively

---

**Algorithm 1** Dataset Construction Process.

**Input:** iteration $\mathcal{N}$, raw schema $\mathcal{S}_{raw}$, and raw dataset $\{\mathcal{D}_{train}^{raw}, \mathcal{D}_{dev}^{raw}, \mathcal{D}_{test}^{raw}\}$
**Output:** Schema $\mathcal{S}_{\mathcal{N}}, \{\mathcal{D}_{train}^{\mathcal{N}}, \mathcal{D}_{dev}^{\mathcal{N}}, \mathcal{D}_{test}^{\mathcal{N}}\}$

1: Randomly initialize $n_{init}$ nodes in $\mathcal{S}_{raw}$ as $\mathcal{S}_1$
2: Pick out the instance associated with schema $\mathcal{S}$ as $\mathcal{D}^{(1)} = \{\mathcal{D}_{train}^1, \mathcal{D}_{dev}^1, \mathcal{D}_{test}^1\}$
3: **for** iteration $i = 2, \ldots, n$ **do**
4:     **Horizontal Schema Expansion:** Compute $\phi_{neighbor}(\vec{W}_c, \vec{W}_s)$ for candidate schema $\mathcal{S}$
5:     **Vertical Schma Expansion:** Select $n_{iter}$ sub node whose *father node* belongs to $\mathcal{S}$ and update
6:     **Hybrid Schema Expansion:** Combine extension Steps 4 and 5
7:     Ouput iteration $i$ dataset schema $\mathcal{S}_i = \mathcal{S}$, instance $\mathcal{D}^{(i)} = \{\mathcal{D}_{dev}^i, \mathcal{D}_{test}^i\}$
8: **end for**

---

adjusted according to the schema. We detail several vanilla baselines as follows and introduce the proposed ADAKGC in §3.

**Vanilla Baselines:** Schema-adaptable KGC can be thought of as a structured prediction language task that transfers information between class nodes through the generalizability of the structure. **TANL** (Paolini et al., 2021) introduces an augmented natural language translation task from which information related to the schema can be implicitly extracted. **TEXT2EVENT** (Lu et al., 2021) is a unified sequence-to-structure architecture for event extraction with a constrained decoding algorithm for event schema knowledge injection during in-

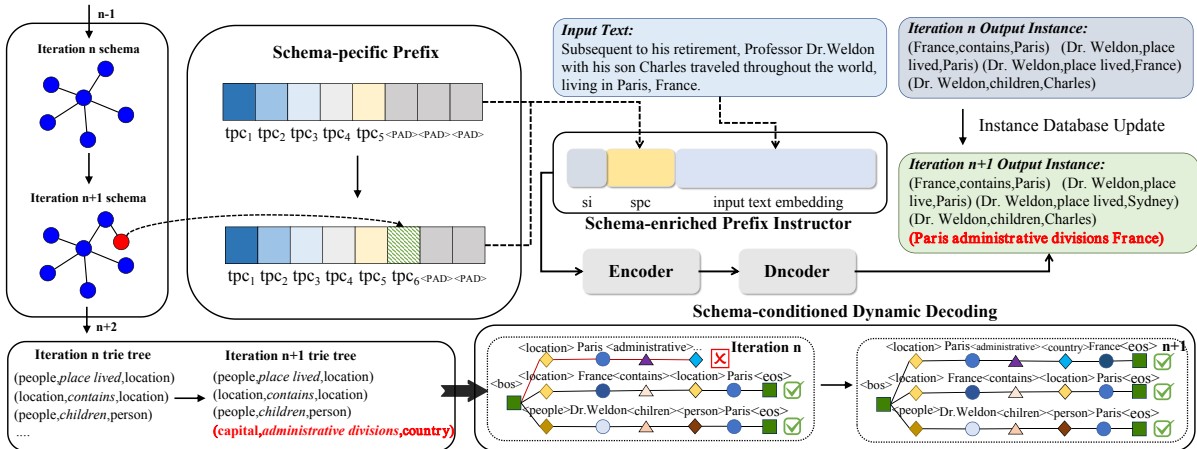

Figure 3: The overview of **ADA**ptive **K**nowledge **G**raph **C**onstruction ADAKGC.

ference. **UIE** (Lu et al., 2022) is a unified text-to-structure generation framework that enables unified modeling of different IE tasks and adaptively generates target sequences by a schema-based prompting mechanism. **GPT-3** (Brown et al., 2020), a large-scale language model (LLM), can serve as a baseline for schema-adaptable KGC. Although current works focusing on structured extraction can achieve excellent performance with static types of knowledge, they are typically unaware of schema evolution. To clarify this issue, we introduce a simple yet effective baseline dubbed schema-**ADA**ptive **K**nowledge **G**raph **C**onstruction ADAKGC.

## 3 The Proposed Baseline: ADAKGC

### 3.1 Overview

As shown in Figure 3, ADAKGC utilizes a pre-trained encoder-decoder language model (LM) T5 (Raffel et al., 2020a) as the basic architecture for the schema-adaptable KGC task. Specifically, let encoder input $X_{en} = [S; X]$ be the concatenation of schema $S$ and input $X$. In the decoding process, the LM calculates the conditioned probability of generating a new token $y_t$ given the previous token $y_{<t}$:

$$p(Y_{de} \mid X_{en}) = \prod_{t=1}^{|Y|} p\left(y_t \mid y_{<t}, S, X\right) \quad (1)$$

We initialize the model using the pre-trained parameter $\theta$. Here, $p_\theta$ is a trainable language model distribution. In the $k$-th iteration, we perform a gradient update on the following log-likelihood objective:

$$\max_\theta \log p_\theta(y \mid x; s_k)$$
$$= \max_\theta \sum_{t \in Y_{index}} \log p_\theta(h_t \mid h_{<t}) \quad (2)$$

where $h_t$ is the activation vector at decoding time step $t$. $h_t = \left[h_t^{(1)}; \cdots ; h_t^{(m)}\right]$ is a concatenation of all activation layers, and $h_t^{(j)}$ is the activation vector of the $j$-th layer at time step $t$.

### 3.2 Schema-enriched Prefix Instructor

Inspired by prefix-tuning (Li and Liang, 2021), we use task-specific prefix instructors to indicate task information, which are pairs of transformer-activated differentiable sequences $\{si_{en}, si_{de}\}$, each containing $p$ consecutive $D$-dim vectors for encoder and decoder. Since using a discrete natural language task instruction in the context (e.g., "The schema used for the task is:") may guide the LM to produce a sub-optimal generated sequence, we optimize the instructions as a continuous soft prompt, propagating upward to all transformer activation layers and rightward to subsequent tokens.

Due to schema changes with iterations, we present schema-specific prefix instructors to instruct the encoding process. Specifically, we formalize the schema graph as linearized text. Assume given the constrained schema of RE task $s_k = \{(h_1, r_1, t_1), ...(h_n, r_n, t_n)\}$ and $tpc_i = (h_i, r_i, t_i)$ denotes the $i$-th triple prefix constraint. By concatenating these schema prefix constraints initialized by word embedding, $spc$ can be dynamically adjusted as the schema evolves, and added padding tokens to be a fixed length when instructing the LM:

$$spc = \text{Concat}\left(tpc_1, \ldots, tpc_n, PAD\right) \quad (3)$$

Thus, the schema-enriched prefix instructor provides a two-part prefix combination $Z = \{si_{en}, spc_c; si_{de}\}$, where ";" separates the respective prefix instructors for encoder and decoder. We recursively activate the decoder transformer activation vector $h_t$, which is the connection of all layers, at time step $t$ in the LM.

$$h_t = \begin{cases} si_t, & \text{if } t \le p \\ \text{LM}\left(y_t, h_{<t} \mid S, X\right), & \text{otherwise} \end{cases} \quad (4)$$

The training parameters of our model contain the LM parameters $\theta$, the encoder-decoder task-specific prefix instructor $\{si_{en}, si_{de}\}$, and the schema-specific constraint instructor $spc$. For stable optimization, we follow Li and Liang (2021) to reparameterize the matrix $M_\phi[t, :] = MLP_\phi(M'_\phi[t, :])$ with a smaller matrix $M'_\phi$ consisting of a large feedforward neural network $MLP_\phi$, which can alleviate the optimization instability caused by directly updating the prefix parameters and is applied to $\{si_{en}, si_{de}; spc\}$. We train the parameters of the model in the following steps: (1) First, freeze other parameters, fine-tune the prefix instructor $\{si_{en}, si_{de}\}$ to learn task-specific prompts; (2) Secondly, freeze $\{si_{en}, si_{de}\}$, optimize the schema-specific instructor $spc$ for the given schema graph; (3) Finally, we unfreeze the LM parameter $\theta$ and collaboratively optimize all parameters to capture the association between the prefix instructor and model parameters.

### 3.3 Schema-conditioned Dynamic Decoding

Previous works leverage a greedy decoding algorithm to generate linearized instance predictions token by token for the hidden sequence $h_t$, which selects the token with the highest prediction probability $p\left(y_t \mid y_{<t}, S, X\right)$ at each vanilla decoding step $t$. Unfortunately, when the schema changes, this decoding algorithm does not guarantee the generated instances are consistent with the latest schema. In other words, it may result in out-of-date or invalid types being generated due to the lack of labeled data fine-tuning the model to adapt the probability distribution to the current schema constraints. In addition, the greedy decoding algorithm neglects useful schema knowledge that can effectively guide the decoding process.

In the schema-conditioned decoding process, we apply a trie-based decoding mechanism that dynamically constructs a trie-tree by leveraging the latest schema. An intuitive interpretation is that the schema contains rich semantic information (*e.g.*, instance types) and structural information (*e.g.*, relational edges between instance types) so that the decoding process can be constrained to ensure that the generated token is valid. Specifically, we constrain the model to generate the type tokens consistently with the existing schema at the type location. We pursue the LM output to be a sequence of RE following pattern and optimized using the standard seq2seq objective function:

$$[bos] \ldots \mathcal{T}_h^{(n)}, \mathcal{E}_h^{(n)}, \mathcal{R}^{(n)}, \mathcal{T}_t^{(n)}, \mathcal{E}_t^{(n)} \ldots [eos]$$

where $\mathcal{E}_h^{(n)}$, $\mathcal{T}_h^{(n)}$, $\mathcal{E}_t^{(n)}$, $\mathcal{E}_h^{(n)}$ refer to the $n$-th generated head entity, tail entity, and their respective types while $\mathcal{R}^{(n)}$ refer to relation.

## 4 Experiments

### 4.1 Experimental Settings

**Datasets.** We conduct experiments on KGC tasks, including NER, RE and EE. The used datasets includes FEW-NERD (Ding et al., 2021), NYT (Riedel et al., 2010b) and ACE2005 (Walker et al., 2006). In our work, we need to construct schema as well as golden validation/test sets dynamically. For each dataset, we build three types of evaluation settings based on §2.3. Therefore for original datasets, we use a certain proportion of the schema as initialization to conduct schema expansion regarding three schema evolution categories in Appendix A.1.
**Evaluation.** We use span-based Micro-F1 as the primary metric. **Rel-S** means that the relation is correct if the relation type is correct and the string and entity types of the related entity mentions are correct. For each iteration experiment, we report the average performance over 3 random seeds. UIE is implemented without pre-training by directly using T5-v1.1-base as the backbone for a fair comparison. More details are in Appendix A.2.

### 4.2 Main Results

We report empirical results regarding horizontal schema expansion, vertical schema expansion and hybrid schema expansion settings to compare our proposed methods with the baselines. The performance over all iterations during the whole schema-adaptable KGC process is presented in Table 1-3. From the results, we can observe that:

| | Model | Iter 1 | Iter 2 | Iter 3 | Iter 4 | Iter 5 | Iter 6 | Iter 7 | AVE |
|---|---|---|---|---|---|---|---|---|---|
| **Entity** (NERD) **Ent-F1** | TANL | 71.52 | 65.21 | 60.38 | 56.37 | 53.21 | 49.66 | 46.55 | 57.56 |
| | UIE | 72.72 | 66.78 | 62.24 | 58.29 | 55.08 | 51.42 | 48.04 | 59.22 |
| | AdaKGC | 72.91 | 66.95 | 62.37 | 58.51 | 55.38 | 51.81 | 48.54 | **59.50** |
| **Relation** (NYT) **Rel-S F1** | TANL | 89.92 | 81.81 | 78.34 | 72.71 | 68.42 | 65.19 | 62.94 | 74.19 |
| | UIE | 90.17 | 82.09 | 78.74 | 73.12 | 69.03 | 65.53 | 63.30 | 74.57 |
| | AdaKGC | 90.34 | 82.33 | 79.03 | 73.34 | 69.19 | 65.87 | 63.58 | **74.81** |
| **Event Trigger** (ACE2005) **Evt Tri F1** | TEXT2EVENT | 69.23 | 68.05 | 65.45 | 61.37 | 60.15 | 59.34 | 54.42 | 62.57 |
| | UIE | 70.75 | 69.13 | 66.20 | 62.19 | 60.90 | 59.83 | 54.74 | 63.39 |
| | AdaKGC | 72.43 | 70.90 | 68.14 | 63.49 | 61.97 | 61.33 | 55.73 | **64.86** |
| **Event Argument** (ACE2005) **Evt Arg F1** | TEXT2EVENT | 46.15 | 44.40 | 42.58 | 39.73 | 39.17 | 38.77 | 35.31 | 40.87 |
| | UIE | 49.14 | 47.90 | 45.75 | 42.32 | 41.83 | 41.27 | 37.60 | 43.69 |
| | AdaKGC | 49.18 | 48.14 | 47.08 | 43.85 | 43.17 | 43.10 | 38.79 | **44.76** |

Table 1: Horizontal schema expansion results in schema-adaptable knowledge graph construction.

| | Model | Iter 1 | Iter 2 | Iter 3 | Iter 4 | Iter 5 | Iter 6 | Iter 7 | AVE |
|---|---|---|---|---|---|---|---|---|---|
| **Entity** (NERD) **Ent-F1** | TANL | 73.49 | 64.83 | 57.05 | 51.84 | 47.16 | 42.06 | 35.96 | 53.20 |
| | UIE | 74.45 | 66.09 | 58.24 | 53.01 | 48.24 | 45.77 | 48.41 | 56.32 |
| | AdaKGC | 74.45 | 66.05 | 58.26 | 53.06 | 48.39 | 45.92 | 48.57 | **56.39** |
| **Relation** (NYT) **Rel-S F1** | TANL | 90.13 | 83.11 | 76.63 | 73.03 | 67.03 | 63.40 | 47.13 | 71.49 |
| | UIE | 90.38 | 81.76 | 74.27 | 71.61 | 65.30 | 62.95 | 59.81 | **72.30** |
| | AdaKGC | 90.18 | 81.65 | 74.13 | 71.48 | 65.10 | 62.75 | 59.77 | 72.15 |
| **Event Trigger** (ACE2005) **Evt Tri F1** | TEXT2EVENT | 67.10 | 55.81 | 50.38 | 52.99 | 45.82 | 41.39 | 41.65 | 50.73 |
| | UIE | 70.94 | 60.00 | 57.01 | 62.52 | 60.39 | 54.90 | 53.62 | 59.91 |
| | AdaKGC | 70.57 | 59.75 | 56.50 | 62.31 | 60.57 | 55.49 | 54.32 | **59.93** |
| **Event Argument** (ACE2005) **Evt Arg F1** | TEXT2EVENT | 49.32 | 37.83 | 33.43 | 35.49 | 30.67 | 27.49 | 27.79 | 34.57 |
| | UIE | 50.70 | 41.45 | 39.66 | 44.14 | 43.65 | 38.86 | 37.77 | 42.32 |
| | AdaKGC | 51.87 | 42.74 | 40.68 | 45.16 | 43.97 | 40.10 | 39.05 | **43.37** |

Table 2: Vertical schema expansion results in schema-adaptable knowledge graph construction.

**Schema adaptive generalization challenge.** On all three expansion categories, the model performances tend to decrease as the iterations increase. TANL achieves lower performance which employs an augmented language and implicitly trains the model to learn schema information. TEXT2EVENT utilizes schema as constraint information on the decoding side and outperforms other models in some iterations. Although ADAKGC and UIE obtain optimal or suboptimal performance, the performance of iteration 1 and iteration 7 has a significant drop. We believe that the implicit schema evolution rules can help future work to develop adaptive generalization capabilities for schema-adaptable KGC.

**Schema-enhanced modules boost the performances.** Compared to other models, ADAKGC is improved with schema-enhanced modules on both the encoder and decoder, which allows it to achieve the best performance in most settings. On the ACE2005 hybrid schema expansion dataset,

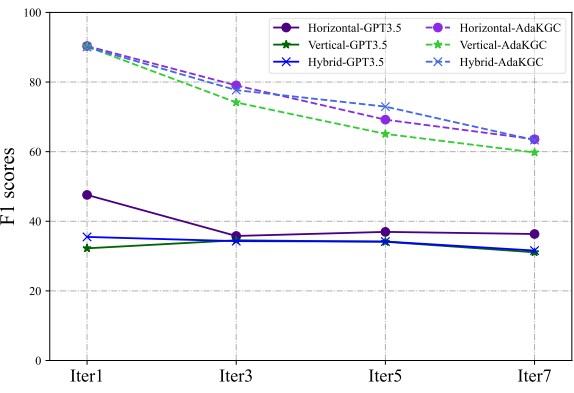

Figure 4: GPT-3.5 results on schema expansion dataset.

ADAKGC improves 0.71% on trigger extraction and 3.65% on event argument extraction, indicating that ADAKGC can capture schema-specific information under evolutionary schema.

**LLMs can understand schema adaption patterns better.** To explore the performance of LLMs (Qiao et al., 2023) on the proposed tasks, we per-

|  | Model | Iter 1 | Iter 2 | Iter 3 | Iter 4 | Iter 5 | Iter 6 | Iter 7 | AVE |
|---|---|---|---|---|---|---|---|---|---|
| **Entity** (**NERD**) **Ent-F1** | TANL | 68.37 | 58.50 | 54.51 | 52.46 | 48.67 | 44.13 | 41.51 | 52.59 |
| | UIE | 69.26 | 59.80 | 55.71 | 53.66 | 49.55 | 46.97 | 48.11 | 54.72 |
| | AdaKGC | 69.48 | 59.97 | 55.94 | 53.89 | 49.92 | 47.44 | 48.43 | **55.01** |
| **Relation** (**NYT**) **Rel-S F1** | TANL | 88.67 | 81.59 | 76.48 | 72.22 | 72.35 | 61.76 | 57.87 | 72.99 |
| | UIE | 90.17 | 83.14 | 77.81 | 71.45 | 72.97 | 65.57 | 63.12 | **74.89** |
| | AdaKGC | 90.07 | 83.06 | 77.68 | 71.38 | 72.94 | 65.73 | 63.32 | 74.88 |
| **Event Trigger** (**ACE2005**) **Evt Tri F1** | TEXT2EVENT | 69.26 | 56.99 | 53.32 | 46.03 | 40.44 | 56.86 | 48.84 | 53.11 |
| | UIE | 74.69 | 66.34 | 63.12 | 63.21 | 59.86 | 53.18 | 53.26 | 61.95 |
| | AdaKGC | 74.84 | 66.99 | 63.28 | 63.07 | 60.94 | 54.72 | 54.80 | **62.66** |
| **Event Argument** (**ACE2005**) **Evt Arg F1** | TEXT2EVENT | 50.32 | 38.14 | 35.79 | 31.94 | 28.79 | 37.75 | 33.54 | 36.61 |
| | UIE | 51.94 | 45.22 | 42.97 | 43.12 | 40.66 | 36.66 | 36.17 | 42.39 |
| | AdaKGC | 55.08 | 48.58 | 46.19 | 46.27 | 45.22 | 40.52 | 40.41 | **46.04** |

Table 3: Hybrid schema expansion results in schema-adaptable knowledge graph construction.

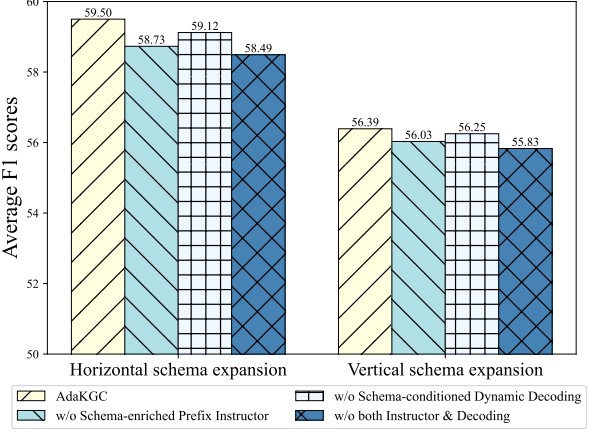

Figure 5: Ablation study on NERD horizontal schema expansion dataset, with the average result of 7 iterations.

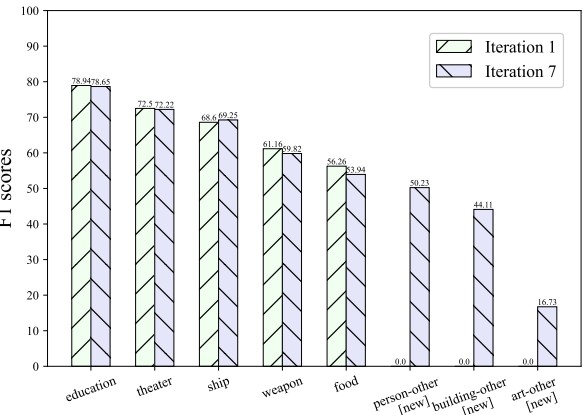

Figure 6: Case study on NERD vertical schema expansion dataset, Iteration 1 vs. Iteration 7.

form comparative experiments with GPT-3.5 on NYT. Since we cannot utilize all training instances, we report in-context learning performance given 20-shot demonstrations as shown in Appendix A.5. From Figure 4, we notice that GPT-3.5 is capable of producing instances that conform to the dynamically changing schema but still yield low performance due to the low-shot issue. Likewise, we sample several cases and use ChatGPT[3] to evaluate schema-adaptable KGC (See Figure 7 and 8 in Appendix A.6), which surprisingly demonstrates stable generalization ability with evolving schema. These findings indicate a promising future work of schema-adaptable KGC to develop alignment prompts with LLMs.

## 4.3 Ablation Study on ADAKGC

To prove the effects of the schema-enriched prefix instructor and schema-conditioned dynamic decod-

ing, we conduct the ablation study, and the results are shown in Figure 5. From two evolutionary categories, we observe that: (1) Both schema-enriched prefix instructor and schema-conditioned dynamic decoding can help the schema-adaptable learning process; (2) Efficiently encoding schema evolution information is more important, which achieves improvements of 0.77% on horizontal schema expansion and 0.36% on vertical schema expansion.

## 4.4 Case Study

As shown in Figure 6, we randomly select 8 types and observe that: (1) The types that appear in the initial schema mostly degrade performance, indicating that the model causes slight confusion as the schema expands. (2) Due to the structural inheritance relationship in the vertical expansion of the schema, our model can effectively transfer the labels of the father node to the child nodes when new child nodes are added.

To further analyze the drawbacks of our model

[3] https://openai.com/blog/chatgpt/

| Error Analysis | Input Example | Gold Type(s) | Predicted Type(s) | Proportion |
|---|---|---|---|---|
| **Weak Transfer** | Kelly, who declined to talk to reporters here, travel to Tokyo Sunday for talks with Japanese officials. | Meet[travel] | Transport[travel] | 38% |
| **Inheritance Deficiency** | He started his entertainment career at ABC, where he is credited with creating the 'movie of the week' concept. | Start-Position[started] | Personnel[started] | 24% |
| **Relevance Neglect** | Kelly, the US assistant secretary for East Asia and Pacific Affairs, arrived in Seoul from Beijing Friday to brief Yoon, the foreign minister. | Transport[arrived] Meet[brief] | Transport[arrived] | 21% |
| **Class Imbalance** | Within weeks he was arrested and charged with sodomising an official driver several years previously and with abusing his powers to cover up the offence. | Arrest-Jail[arrested] Charge-Indict[charged] | Transport[arrested] | 13% |
| **Potential Annotation** | Anne-Marie will get the couple's 19-room home in New York state, which was on the market last year for 21.5 million dollars, as well as their fine art collection. | None Event | Transfer-ownership[get] | 4% |

Table 4: Error analysis on all ACE2005 schema expansion datasets.

and promote future works of schema-adaptable KGC, we count incorrect instances and classify them into five categories below, as shown in Table 4: (1) *Weak Transfer.* Despite schema expansion, the model is prevented from updating labels by old model parameters. (2) *Inheritance Deficiency.* The label is not inherited in time when subdividing the father node. (3) *Relevance Neglect.* The lack of ontology relevance leads to the absence of correlated event extraction. (4) *Class Imbalance.* Models suffering from unbalanced class learning problems tend to depend on similarly in-context sentences to judge high-frequency labels. (5) *Potential Annotation.* Some example outputs suggest potential errors or omitted annotation.

## 5   Related Work

### 5.1   Knowledge Graph Construction

Automatic construction of knowledge graphs from textual or structured data has attracted extensive research in recent years, including tasks such as NER (Paolini et al., 2021; Cui et al., 2021a), RE (Lin et al., 2020; Joshi et al., 2020; Ye et al., 2021), EE (Ramponi et al., 2020; Liu et al., 2020; Lu et al., 2021), etc. In contrast to closed-domain knowledge extraction, open knowledge extraction (Kolluru et al., 2020; Zhan and Zhao, 2020; Kotnis et al., 2022; Wang et al., 2022b) is oriented toward the absence of schema constraints and can quickly generate extensive and meaningful knowledge. However, the ignoring of schema introduces uncertainty and ambiguity in output control, and we believe that a clear setting can be chosen to track the realignment of instances. Besides, KGC in low-resource scenarios (Huang et al., 2018; Zhang et al., 2020; Schick and Schütze, 2021; Chen et al., 2022b; Ye et al., 2022; Zhang et al., 2022c) requires the model to predict new instances with only lim-

ited training instances available. As opposed to this instance-driven KGC approach, we argue that the schema-driven approach can leverage evolutionary instructions provided with richer ontological associations, resulting in new challenges and research opportunities.

### 5.2   Lifelong Learning

Lifelong learning is aimed at training new classes online without catastrophic forgetting. Generally, lifelong learning mainly falls into four categories: regularization-based (Kirkpatrick et al., 2017; Zenke et al., 2017; Aljundi et al., 2018), replay-based (Lopez-Paz and Ranzato, 2017; Shin et al., 2017), architecture-based (Mallya and Lazebnik, 2018; Yoon et al., 2018) and knowledge distillation (Chuang et al., 2020; Cao et al., 2020). To study class-incremental learning, Monaikul et al. (2021) builds a unified NER classifier for all the classes encountered over time, while Wang et al. (2022a) develops a framework to reconstruct synthetic training data of the old classes. Recently, (Wang et al., 2019) proposes a lifelong RE method that employs an explicit alignment model to overcome forgetting, while (Shen et al., 2020) presents a self-adaptive dynamic regularization method. To address class incremental learning in event detection (Cao et al., 2020), Yu et al. (2021) takes advantage of rich correlations among ontology types, and Liu et al. (2022) adopts continuous prompts to learn event-specific representation for prediction. Compared with previous work that focused only on class increments, we discuss three principles of schema expansion from the potential demand for schema adaptation.

## 6 Conclusion

This paper introduces a new task of schema-adaptable KGC with benchmark datasets and a new baseline ADAKGC. We illustrate the task difficulties with previous baselines on three principles of schema expansion patterns (horizontal, vertical, hybrid) and demonstrate the effectiveness of the proposed ADAKGC.

## Acknowledgment

We would like to express gratitude to the anonymous reviewers for their kind comments. This work was supported by the National Natural Science Foundation of China (No.62206246), Zhejiang Provincial Natural Science Foundation of China (No. LGG22F030011), Ningbo Natural Science Foundation (2021J190), Yongjiang Talent Introduction Programme (2021A-156-G), CCF-Baidu Open Fund, and Information Technology Center and State Key Lab of CAD&CG, Zhejiang University.

## Limitations

The proposed work still contains several limitations, as follows:

**Datasets:** Note that several datasets, such as ACE2005, cannot be released due to LICENCE issues; we release the code to build the datasets and provide all the pre-processed publicly available datasets (e.g., Few-NERD, NYT) We use several existing datasets to construct schema-adaptable benchmarks; however, previous datasets may have limited schema structures (the schema pattern in some datasets is very simple). We plan to build more datasets via crowdsourcing for comprehensive evaluation. In addition, we will continue to promote the construction of multimodal schema adaptive graphs, which leverage the dynamic evolution of schema to integrate visual and textual knowledge into a self-learning graph extraction system.

**Baselines and Proposed ADAKGC:** Note that the proposed one, although better than previous approaches, including Text2Event (Lu et al., 2021), TANL (Paolini et al., 2021), UIE (Lu et al., 2022), still suffers from poor generalization ability. However, we notice a very stable performance with LLM (though deficient performance), indicating a new promising solution for schema-adatable KGC.

## Ethical Considerations

**Intended use.** The dataset and model in this paper are indented to be used for exploratory analysis of schema-adaptable KGC.

**Biases.** We collect data from existing datasets (e.g., Few-NERD: CC BY-SA 4.0 license.), which may contain some data with offensive language or discriminatory.

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

# A  Appendix

This section describes the details of experiments, including dataset construction and evaluation on downstream tasks.

## A.1  Dataset Construction

### A.1.1  Construction Process

In each task, we execute three schema evolution strategies. The raw dataset statistics are shown in Table 5, where it can be seen that they have a two-level schema structure, leaving the research of a more hierarchical schema structure for future work. As shown in Algorithm 2-4, we describe in detail the specific construction process of horizontal schema extension, vertical schema extension and hybrid schema extension.

In particular, we also release additional datasets from a semantic substitution perspective. As shown in Algorithm 5, analogous schema expansion requires schema replacement for semantically similar new nodes. Based on the performance of the old class transfer to the new semantic class, we can evaluate the semantic invariance capability.

| | #Maj | #Sub | #Train | #Val | #Test |
|---|---|---|---|---|---|
| NERD | 8 | 66 | 131,767 | 18,824 | 37,648 |
| NYT | 4 | 24 | 56,196 | 5,000 | 5,000 |
| ACE-2005 | 8 | 33 | 19,216 | 901 | 676 |

Table 5: Raw datasets statistics. #Maj indicates the number of major classes, #Sub is the number of subclasses, and #Val #Test is the number of sentences.

**Horizontal Schema Expansion.** Neighboring nodes of the specified type that have high-level

similarity values in the same framework are also adjacent when projected into the semantic space (Huang et al., 2018). Existing research efforts have developed many rich libraries of ontologies (*e.g.* FrameNet (Baker and Sato, 2003), VerbNet (Kipper et al., 2008), Propbank (Palmer et al., 2005), and OntoNotes (Pradhan et al., 2007)), where each ontology type is associated with a set of pre-defined neighboring ontologies. (1) Searching the ontology library to retrieve candidate nodes $W_f$ associated with target nodes $W_s$ at the same hierarchy. (2) The similarity metric is obtained by calculating the cosine vector similarity of all candidate nodes $W_f$ to the target node $W_s$ (Eq. 5). (3) Selecting the appropriate threshold of node pairs to confirm the sorted addition of horizontal nodes. (4) Updating the schema with horizontal nodes and adding the golden validation set and test set in the dataset.

$$\phi_{Sim}(\vec{W}_f, \vec{W}_s) = \frac{\sum_{i=1}^{|W_s|} w_i \cdot w_f}{\|\vec{W}_f\|_2 \cdot \|\vec{W}_s\|_2} \quad (5)$$

**Vertical Schema Expansion.** Structural similarity needs to be exploited when adding schema hierarchy nodes as new classes. (1) For search convenience, we link the hypernym ontology under a root node so that the schema forms a tree structure. (2) Starting at the root node, we utilize a child selection strategy by recursively applying through the tree until reaching the deepest node. A node could be expandable when it represents a non-terminal state or has hyponyms in semantics (e.g., location->country). (3) According to the available hyponyms, one (or more) child nodes are added to expand the current schema tree. (4) Updating the schema with vertical nodes and adding the golden validation set and test set in the dataset.
**Hybrid Schema Expansion.** It is necessary to hybrid horizontal and vertical expansion to form a comprehensive structural topology, which is more consistent with real scenarios. (1) Setting the threshold $\alpha$ for random selection. (2) Executing horizontal expansion iteration below the threshold $\alpha$, or vertical node expansion above the threshold $\alpha$. Note that when the father node of added nodes does not exist, we also add the father node to maintain the schema hierarchy. (3) Updating the schema with the corresponding nodes and adding the golden validation set and test set in the dataset.
**Analogous Schema Expansion.** To detect the semantic node sensitivity of the schema, we randomly

replace similar semantic expressions for the nodes. (1) Random selection of candidate nodes to obtain word expressions $W_C$. (2) Candidate nodes are created by pairing $W_C$ with all words in the corpus word list $W_L$. The consistency between individual words is calculated by the normalized point-by-point mutual information (NPMI) of $w_i$ and $w_j$ (Eq. 6), where adding smooth $\epsilon$ and $\gamma$ controls for $\log p(w_i, w_j)$ weights for higher NPMI values (Eq. 7). (3) Adopting candidate nodes that exceed the threshold to replace the schema and updating the golden validation set and test set in the dataset.

$$\vec{v}(W) = \left\{ \sum_{w_i \in W_C} NMPI(w_i, w_j)^{\gamma} \right\}_{j=1,\dots,|W_L|} \quad (6)$$

$$NMPI(w_i, w_j)^{\gamma} = \left( \frac{\log \frac{p(w_i, w_j) + \epsilon}{p(w_i) \cdot p(w_j)}}{-\log p(w_i, w_j) + \epsilon} \right)^{\gamma} \quad (7)$$

### A.1.2 Schema-adaptable Datasets Statistic

We set the number $\mathcal{N}$ of total iterations, and initialize the original number of schema nodes. We show the statistics of schema-adaptable datasets for each task in Table 6.

### A.2 Evaluation

We use span-based Micro-F1 as the major metric to evaluate the model and adopt the same evaluation metrics as previous work:

* **Named Entity Recognition**: an entity mention is correct if its strings and type match a reference entity.

* **Relation Strict**: a relation is correct if its relation type is correct and the string and entity types of the related entity mentions are correct.

* **Event Trigger**: an event trigger is correct if its strings and event type match a reference trigger.

* **Event Argument**: an event argument is correct if its strings, role type, and event type match a reference argument mentioned.

| Taxonomy | Horizontal | | | Vertical | | | Hybrid | | | Analogous | | |
|---|---|---|---|---|---|---|---|---|---|---|---|---|
| | #Init | #Add | #N | #Init | #Add | #N | #Init | #Add | #N | #Init | #Add | #N |
| NERD | 30 | 6 | 7 | 30 | 6 | 7 | 30 | 6 | 7 | 66 | - | 7 |
| NYT | 10 | 2 | 8 | 10 | 2 | 8 | 10 | 2 | 8 | 24 | - | 7 |
| ACE-2005 | 15 | 3 | 7 | 15 | 3 | 7 | 15 | 3 | 7 | 33 | - | 7 |

Table 6: Schema-adaptable datasets statistics. #Init indicates the number of initial subclasses, #Add is the number of subclasses added per iteration, and #N is the total number of iterations.

## A.3 Hyper-parameters

We adopt T5-v1.1-base (Raffel et al., 2020b), which has 12 layers of the encoder, 12 layers of the decoder, 768 hidden units, and 12 attention heads as the backbone. Specifically, we utilize Pytorch (Paszke et al., 2017) to conduct experiments with batch size 16 on one NVIDIA 3090 GPU. We detail the hyper-parameters for each dataset as follows:

**NERD.** The hyper-parameter search space is:

- epoch: 15

- batch size: 16

- accumulate: 1

- learning rate: [**1e-4**, 3e-4, 5e-4]

- warmup rate: 0.06

**NYT.** The hyper-parameter search space is:

- epoch: 20

- batch size: 16

- accumulate: 1

- learning rate: [**1e-4**, 3e-4, 5e-4]

- warmup rate: 0.06

**ACE2005.** The hyper-parameter search space is:

- epoch: 30

- batch size: 16

- accumulate: 1

- learning rate: [**1e-4**, 3e-4, 5e-4]

- warmup rate: 0.06

## A.4 Analogous Schema Expansion Experiment

As shown in Figure 7, our ADAKGC also has powerful semantic transplantation capabilities, which achieves competitive performance with baselines. With the schema-enriched prefix instructor, ADAKGC achieves an improvement of 7.70% on average over TEXT2EVENT on the event trigger extraction task and 4.87% on the event argument extraction task. This verifies the proposed schema-enriched prefix instructor and decoding modules can learn general schema-adaptable ability even the schema evolution knowledge is rarely in the pre-training stage. Note that TANL achieves the best performance on the NYT dataset, indicating that language models have the ability to learn schema semantic transfer implicitly as an augmented natural language prediction task. Therefore we believe that in addition to the schema structure perception modules, semantic robustness modules for analogous node expansion scenarios are also essential.

## A.5 GPT-3.5 Experiment Details

GPT-3.5 is a large autoregressive language model with 175 billion parameters. To explore the performance of GPT-3.5 on the schema-adaptive KGC task, we follow the input format of few-shot learning using OpenAI API[4]. As shown in Table 12, we utilize a fixed manual template to generate a contextual window suitable for the model, including natural language task descriptions (text in blue), linearized schemas (text in purple), 20 examples in the model's context, and task prompts (text in red).

## A.6 ChatGPT Results

ChatGPT[5] trains an initial model using supervised fine-tuning and further utilizes reinforcement learning systems to rank by quality for human feedback rewards. We handle Schema-adaptable KGC tasks

---

[4]https://platform.openai.com/docs/models/gpt-3-5
[5]https://chat.openai.com/chat

**Algorithm 2** The construction process of horizontal schema expansion.

---
1: Set sampling seed $\theta$, total iteration $N$, raw schema $S_{raw}$, and raw dataset $\{D_{train}^{raw}, D_{dev}^{raw}, D_{test}^{raw}\}$
2: Initialize blank schema $S$, blank dataset $\{D_{train}, D_{dev}, D_{test}\}$ and initial node number $n_{init}$, node number $n_{iter}$ per iteration
3: Randomly select $n_{init}$ nodes in $S_{raw}$, $S = S \cup$ init node, $S_1 = S$, $S_{raw} = S_{raw} - S$
4: Pick out the instance associated with $S$, $\{D_{train}^1, D_{dev}^1, D_{test}^1\} = \{D_{train}, D_{dev}, D_{test}\}$
5: **for** iteration $N$ **do**
6:    Calculate $\phi_{Sim}(\vec{W_f}, \vec{W_s}) = \frac{\sum_{i=1}^{|W_s|} w_i \cdot w_f}{\|\vec{W_f}\|_2 \cdot \|\vec{W_s}\|_2}$ for node between $S_{raw}$ and $S$
7:    Pick out top $n_{iter}$ schema node, $S = S \cup S_{raw}[: n_{iter}]$, $S_{raw} = S_{raw} - S_{raw}[: n_{iter}]$
8:    Iteration $i$ dataset $S_i = S$, $\{D_{dev}^i, D_{test}^i\} = \{D_{dev}, D_{test}\}$
9: **end for**

---

by asking questions to the chatbot in a conversational mode. First, we present the task description and the 20 demonstrations as shown in Figure 7. Then we give a paragraph text to test whether the chatbot can extract the corresponding triples based on the same schema as the demonstration examples comply with. From Figure 8 we can find that some of the facts are well extracted, indicating that ChatGPT can understand the task and perform extraction consistent with the schema. Finally, we add three new nodes *"profession" "place founded" "founders"* to the previous schema under a horizontal schema expansion iteration. Output results in Figure 8 show that ChatGPT not only adapts the output to the updated schema but also deduces reasonable facts by a chain-of-thought approach.

**Algorithm 3** The construction process of vertical schema expansion.

---
1: Set sampling seed $\theta$, total iteration $N$, raw schema $S_{raw}$, and raw dataset $\{D_{train}^{raw}, D_{dev}^{raw}, D_{test}^{raw}\}$
2: Initialize blank schema $S$, blank dataset $\{D_{train}, D_{dev}, D_{test}\}$ and initial node number $n_{init}$, node number $n_{iter}$ per iteration
3: **for** major node in $S_{raw}$ **do**
4:    $S = S \cup$ major node
5: **end for**
6: Randomly select $n_{init}$ nodes in $S_{raw}$, $S = S \cup$ init node, $S_1 = S$, $S_{raw} = S_{raw} - S$
7: Pick out the instance associated with $S$, $\{D_{train}^1, D_{dev}^1, D_{test}^1\} = \{D_{train}, D_{dev}, D_{test}\}$
8: **for** iteration $N$ **do**
9:    Randomly select $n_{iter}$ sub node, whose parent belongs to $S$
10:    $S = S \cup S_{raw}[: n_{iter}]$, $S_{raw} = S_{raw} - S_{raw}[: n_{iter}]$
11:    Iteration $i$ dataset $S_i = S$, $\{D_{dev}^i, D_{test}^i\} = \{D_{dev}, D_{test}\}$
12: **end for**

---

**Algorithm 4** The construction process of hybrid schema expansion.

---

1: Set sampling seed $\theta$, hybrid ratio $\alpha$, total iteration $N$, raw schema $S_{raw}$, and raw dataset $\{D_{train}^{raw}, D_{dev}^{raw}, D_{test}^{raw}\}$
2: Initialize blank schema $S$, blank dataset $\{D_{train}, D_{dev}, D_{test}\}$ and initial node number $n_{init}$, node number $n_{iter}$ per iteration
3: Randomly select $n_{init}$ nodes in $S_{raw}$, $S = S \cup$ init node, $S_1 = S$, $S_{raw} = S_{raw} - S$
4: Pick out the instance associated with $S$, $\{D_{train}^1, D_{dev}^1, D_{test}^1\} = \{D_{train}, D_{dev}, D_{test}\}$
5: **for** iteration $N$ **do**
6:    **if** random(0,1)$< \alpha$ **then**
7:       Calculate $\phi_{Sim}(\vec{W}_f, \vec{W}_s) = \frac{\sum_{i=1}^{|W_s|} w_i \cdot w_f}{\|\vec{W}_f\|_2 \cdot \|\vec{W}_s\|_2}$ for node between $S_{raw}$ and $S$
8:       Pick out top $n_{iter}$ schema node, $S = S \cup S_{raw}[: n_{iter}]$, $S_{raw} = S_{raw} - S_{raw}[: n_{iter}]$
9:    **else**
10:      Randomly select $n_{iter}$ sub node, whose parent belongs to $S$
11:      $S = S \cup S_{raw}[: n_{iter}]$, $S_{raw} = S_{raw} - S_{raw}[: n_{iter}]$
12:    **end if**
13:    Iteration $i$ dataset $S_i = S$, $\{D_{dev}^i, D_{test}^i\} = \{D_{dev}, D_{test}\}$
14: **end for**

---

**Algorithm 5** The construction process of analogous schema expansion.

---

1: Set sampling seed $\theta$, total iteration $N$, raw schema $S_{raw}$, and raw dataset $\{D_{train}^{raw}, D_{dev}^{raw}, D_{test}^{raw}\}$
2: Initialize blank schema $S$, blank dataset $\{D_{train}, D_{dev}, D_{test}\}$ and node number $n_{iter}$ per iteration
3: Initialize $S = S_{raw}$, $S_1 = S$, $\{D_{train}, D_{dev}, D_{test}\} = \{D_{train}^{raw}, D_{dev}^{raw}, D_{test}^{raw}\}$
4: Pick out the instance associated with $S$, $\{D_{train}^1, D_{dev}^1, D_{test}^1\} = \{D_{train}, D_{dev}, D_{test}\}$
5: **for** iteration $N$ **do**
6:    Randomly select $n_{iter}$ schema node in $S$, calculate $\vec{v}(W) = \left\{\sum_{w_r \in W} NMPI(w_r, w_j)^\gamma\right\}_{j=1,\dots,|W_L|}$ for each node, create candidate nodes by pairing $W_r$ with all words $W_L$ in the corpus word list
7:    Replace $S[: n_{iter}]$ with candidate nodes with analogous semantics
8:    Iteration $i$ dataset $S_i = S$, $\{D_{dev}^i, D_{test}^i\} = \{D_{dev}, D_{test}\}$
9: **end for**

| | Model | Iter 1 | Iter 2 | Iter 3 | Iter 4 | Iter 5 | Iter 6 | Iter 7 | AVE |
|---|---|---|---|---|---|---|---|---|---|
| **Entity** (**NERD**) **Ent-F1** | TANL | 68.34 | 62.70 | 57.22 | 52.94 | 47.73 | 42.24 | 36.09 | 52.47 |
| | UIE | 67.58 | 63.39 | 60.37 | 58.00 | 58.17 | 54.26 | 49.53 | 58.76 |
| | AdaKGC | 68.22 | 63.87 | 61.01 | 58.18 | 58.76 | 54.58 | 49.81 | **59.20** |
| **Relation** (**NYT**) **Rel-S F1** | TANL | 89.80 | 86.82 | 83.51 | 77.99 | 73.61 | 73.32 | 66.37 | **78.77** |
| | UIE | 89.66 | 86.56 | 83.27 | 77.80 | 73.30 | 73.27 | 66.33 | 78.60 |
| | AdaKGC | 89.17 | 86.12 | 82.76 | 77.33 | 72.85 | 72.91 | 65.84 | 78.14 |
| **Event Trigger** (**ACE2005**) **Evt Tri F1** | TEXT2EVENT | 64.40 | 60.39 | 53.96 | 52.64 | 40.83 | 32.45 | 29.58 | 47.75 |
| | UIE | 69.69 | 64.53 | 62.10 | 58.47 | 49.76 | 42.51 | 39.31 | 55.20 |
| | AdaKGC | 69.63 | 64.77 | 61.86 | 58.31 | 50.08 | 43.07 | 40.40 | **55.45** |
| **Event Argument** (**ACE2005**) **Evt Arg F1** | TEXT2EVENT | 45.88 | 40.96 | 34.21 | 33.40 | 24.49 | 19.62 | 16.77 | 30.76 |
| | UIE | 49.96 | 42.67 | 40.50 | 36.56 | 28.70 | 24.06 | 20.47 | 34.70 |
| | AdaKGC | 51.74 | 44.56 | 42.48 | 32.61 | 30.52 | 25.41 | 22.10 | **35.63** |

Table 7: Analogous schema replacement results in schema-adaptable knowledge graph construction.

| | Labels |
|---|---|
| *Input:* The Belgrade district court said that Markovic will be tried along with 10 other Milosevic-era officials who face similar charges of 'inappropriate use of state property' that carry a sentence of up to five years in jail. | Labels |
| *Iteration 1 schema:* "attack", "start position", "transfer ownership", "be born", "sentence", "die", "arrest jail", "transport", "elect", "phone write", "end organization", "sue", "acquit", "marry", "extradite" | Sentence[sentence] |
| *Iteration 2 schema:* "attack", "start position", "transfer ownership", "be born", "sentence", "die", "arrest jail", "transport", "elect", "injure", "phone write", "fine", "convict", "end organization", "sue", "acquit", "marry", "extradite" | Sentence[sentence] |
| *Iteration 3 schema:* "attack", "start position", "transfer money", "transfer ownership", "be born", "sentence", "die", "demonstrate", "arrest jail", "transport", "elect", "injure", "phone write", "fine", "convict", "end organization", "sue", "acquit", "execute", "marry", "extradite" | Sentence[sentence] |
| *Iteration 4 schema:* "end position", "attack", "start position", "transfer money", "transfer ownership", "be born", "sentence", "die", "demonstrate", "arrest jail", "transport", "elect", "start organization", "injure", "phone write", "fine", "convict", "end organization", "sue", "acquit", "execute", "marry", "extradite", "pardon" | Sentence[sentence] |
| *Iteration 5 schema:* "end position", "attack", "start position", "transfer money", "transfer ownership", "be born", "sentence", "die", "demonstrate", "arrest jail", "transport", "elect", "start organization", "injure", "phone write", "declare bankruptcy", "trial hearing", "fine", "convict", "end organization", "sue", "acquit", "appeal", "execute", "marry", "extradite", "pardon" | Sentence[sentence] Trial hearing[tried] |
| *Iteration 6 schema:* "end position", "attack", "start position", "charge indict", "transfer money", "transfer ownership", "release parole", "be born", "sentence", "die", "demonstrate", "arrest jail", "transport", "elect", "start organization", "injure", "phone write", "merge organization", "declare bankruptcy", "trial hearing", "fine", "convict", "end organization", "sue", "acquit", "appeal", "execute", "marry", "extradite", "pardon" | Sentence[sentence] Trial hearing[tried] Charge-Indict[charges] |
| *Iteration 7 schema:* "end position", "attack", "start position", "nominate", "charge indict", "transfer money", "transfer ownership", "release parole", "be born", "sentence", "die", "demonstrate", "arrest jail", "transport", "elect", "start organization", "meet", "injure", "phone write", "merge organization", "declare bankruptcy", "trial hearing", "fine", "convict", "end organization", "sue", "divorce", "acquit", "appeal", "execute", "marry", "extradite", "pardon" | Sentence[sentence] Trial hearing[tried] Charge-Indict[charges] |

Table 8: Adaptive evolution of horizontal schema expansion on ACE2005 dataset.

| *Input:* Kelly, the US assistant secretary for East Asia and Pacific Affairs, arrived in Seoul from Beijing Friday to brief Yoon, the foreign minister. | Labels |
|---|---|
| *Iteration 1 schema:* "personnel", "attack", "justice", "transfer money", "transfer ownership", "release parole", "be born", "sentence", "die", "demonstrate", "transport", "business", "contact", "life", "fine", "sue", "execute", "marry", "extradite", "pardon" | Transport[arrived] Contact[brief] |
| *Iteration 2 schema:* "personnel", "attack", "justice", "transfer money", "transfer ownership", "release parole", "be born", "sentence", "die", "demonstrate", "transport", "business", "meet", "life", "contact", "fine", "sue", "acquit", "appeal", "execute", "marry", "extradite", "pardon" | Transport[arrived] Meet[brief] |
| *Iteration 3 schema:* "personnel", "attack", "justice", "transfer money", "transfer ownership", "release parole", "be born", "sentence", "die", "demonstrate", "transport", "start organization", "meet", "life", "contact", "merge organization", "business", "trial hearing", "fine", "sue", "acquit", "appeal", "execute", "marry", "extradite", "pardon" | Transport[arrived] Meet[brief] |
| *Iteration 4 schema:* "personnel", "attack", "justice", "transfer money", "transfer ownership", "release parole", "be born", "sentence", "die", "demonstrate", "transport", "start organization", "meet", "life", "phone write", "merge organization", "declare bankruptcy", "trial hearing", "fine", "end organization", "sue", "acquit", "appeal", "execute", "marry", "extradite", "pardon" | Transport[arrived] Meet[brief] |
| *Iteration 5 schema:* "personnel", "attack", "charge indict", "transfer money", "transfer ownership", "release parole", "be born", "sentence", "die", "demonstrate", "arrest jail", "transport", "start organization", "meet", "life", "phone write", "merge organization", "declare bankruptcy", "trial hearing", "fine", "convict", "end organization", "sue", "acquit", "appeal", "execute", "marry", "extradite", "pardon" | Transport[arrived] Meet[brief] |
| *Iteration 6 schema:* "end position", "attack", "personnel", "charge indict", "transfer money", "transfer ownership", "release parole", "be born", "sentence", "die", "demonstrate", "arrest jail", "transport", "start organization", "meet", "injure", "phone write", "merge organization", "declare bankruptcy", "trial hearing", "fine", "convict", "end organization", "sue", "divorce", "acquit", "appeal", "execute", "marry", "extradite", "pardon" | Transport[arrived] Meet[brief] |
| *Iteration 7 schema:* "end position", "attack", "start position", "nominate", "charge indict", "transfer money", "transfer ownership", "release parole", "be born", "sentence", "die", "demonstrate", "arrest jail", "transport", "elect", "start organization", "meet", "injure", "phone write", "merge organization", "declare bankruptcy", "trial hearing", "fine", "convict", "end organization", "sue", "divorce", "acquit", "appeal", "execute", "marry", "extradite", "pardon" | Transport[arrived] Meet[brief] |

Table 9: Adaptive evolution of vertical schema expansion on ACE2005 dataset. Underlined classes refer to major classes, which will be covered by refined sub classes.

| | Labels |
|---|---|
| *Input:* The charismatic leader of Turkey's governing party was named prime minister Tuesday, a step that probably boosts chances that the United States will get permission to deploy troops in the country along Iraq's northern border. | |
| *Iteration 1 schema:* "attack", "justice", "transfer money", "transfer ownership", "release parole", "be born", "sentence", "die", "demonstrate", "transport", "life", "fine", "sue", "execute", "marry", "extradite", "pardon" | Transport[deploy] |
| *Iteration 2 schema:* "attack", "justice", "transfer money", "transfer ownership", "release parole", "be born", "sentence", "die", "demonstrate", "transport", "meet", "life", "contact", "fine", "sue", "acquit", "appeal", "execute", "marry", "extradite", "pardon" | Transport[deploy] |
| *Iteration 3 schema:* "attack", "justice", "transfer money", "transfer ownership", "release parole", "be born", "sentence", "die", "demonstrate", "transport", "start organization", "meet", "life", "contact", "merge organization", "business", "trial hearing", "fine", "sue", "acquit", "appeal", "execute", "marry", "extradite", "pardon" | Transport[deploy] |
| *Iteration 4 schema:* "attack", "justice", "transfer money", "transfer ownership", "release parole", "be born", "sentence", "die", "demonstrate", "transport", "start organization", "meet", "life", "phone write", "merge organization", "declare bankruptcy", "trial hearing", "fine", "end organization", "sue", "acquit", "appeal", "execute", "marry", "extradite", "pardon" | Transport[deploy] |
| *Iteration 5 schema:* "attack", "charge indict", "transfer money", "transfer ownership", "release parole", "be born", "sentence", "die", "demonstrate", "arrest jail", "transport", "start organization", "meet", "life", "phone write", "merge organization", "declare bankruptcy", "trial hearing", "fine", "convict", "end organization", "sue", "acquit", "appeal", "execute", "marry", "extradite", "pardon" | Transport[deploy] |
| *Iteration 6 schema:* "end position", "attack", "personnel", "charge indict", "transfer money", "transfer ownership", "release parole", "be born", "sentence", "die", "demonstrate", "arrest jail", "transport", "start organization", "meet", "injure", "phone write", "merge organization", "declare bankruptcy", "trial hearing", "fine", "convict", "end organization", "sue", "divorce", "acquit", "appeal", "execute", "marry", "extradite", "pardon" | Transport[deploy] Personnel[named] |
| *Iteration 7 schema:* "end position", "attack", "start position", "nominate", "charge indict", "transfer money", "transfer ownership", "release parole", "be born", "sentence", "die", "demonstrate", "arrest jail", "transport", "elect", "start organization", "meet", "injure", "phone write", "merge organization", "declare bankruptcy", "trial hearing", "fine", "convict", "end organization", "sue", "divorce", "acquit", "appeal", "execute", "marry", "extradite", "pardon" | Transport[deploy] Elect[named] |

Table 10: Adaptive evolution of hybrid schema expansion on ACE2005 dataset. Underlined classes refer to father classes, which occurs when directly adding sub classes that corresponding major class not exists.

| Input: Webb also said details of the breakdowns of the Welches' previous marriages were likely to come up , and cited reports of alleged extramarital affairs by both. | Labels |
|---|---|
| *Iteration 1 schema:* "end position", "attack", "start position", "nominate", "charge indict", "transfer money", "transfer ownership", "release parole", "be born", "sentence", "die", "demonstrate", "arrest jail", "transport", "elect", "start organization", "meet", "injure", "phone write", "merge organization", "declare bankruptcy", "trial hearing", "fine", "convict", "end organization", "sue", "divorce", "acquit", "appeal", "execute", "marry", "extradite", "pardon" | Divorce[breakdowns] Marry[marriages] |
| *Iteration 2 schema:* "end position", "attack", "begin", "nominate", "charge indict", "transfer money", "transfer ownership", "release parole", "be born", "sentence", "die", "demonstrate", "arrest jail", "carry", "elect", "start organization", "meet", "injure", "phone write", "merge organization", "declare bankruptcy", "trial hearing", "fine", "convict", "end organization", "sue", "separate", "acquit", "appeal", "execute", "marry", "extradite", "pardon" | Separate[breakdowns] Marry[marriages] |
| *Iteration 3 schema:* "end", "attack", "begin", "nominate", "prosecute", "remittance", "transfer ownership", "release parole", "be born", "sentence", "die", "demonstrate", "arrest jail", "carry", "elect", "start organization", "meet", "injure", "phone write", "merge organization", "declare bankruptcy", "trial hearing", "fine", "convict", "end organization", "sue", "separate", "acquit", "appeal", "execute", "marry", "extradite", "pardon" | Separate[breakdowns] Marry[marriages] |
| *Iteration 4 schema:* "end", "attack", "begin", "nominate", "prosecute", "remittance", "transfer ownership", "release parole", "be born", "sentence", "pass away", "demonstrate", "arrest jail", "carry", "elect", "start organization", "meet", "injure", "phone write", "merge organization", "declare bankruptcy", "attend the trial", "fine", "convict", "end organization", "sue", "separate", "acquit", "appeal", "perform", "marry", "extradite", "pardon" | Separate[breakdowns] Marry[marriages] |
| *Iteration 5 schema:* "end", "attack", "begin", "nominate", "prosecute", "remittance", "transfer ownership", "release parole", "be born", "condemn", "pass away", "demonstrate", "arrest jail", "carry", "elect", "start organization", "encounter", "injure", "phone write", "merge organization", "go out of business", "attend the trial", "fine", "convict", "end organization", "sue", "separate", "acquit", "appeal", "perform", "marry", "extradite", "pardon" | Separate[breakdowns] Marry[marriages] |
| *Iteration 6 schema:* "end", "attack", "begin", "nominate", "prosecute", "remittance", "giveaway", "release parole", "be born", "condemn", "pass away", "parade", "arrest jail", "carry", "vote", "start organization", "encounter", "injure", "phone write", "merge organization", "go out of business", "attend the trial", "fine", "convict", "end organization", "sue", "separate", "acquit", "appeal", "perform", "marry", "extradite", "pardon" | Separate[breakdowns] Marry[marriages] |
| *Iteration 7 schema:* "end", "attack", "begin", "nominate", "prosecute", "remittance", "giveaway", "release parole", "be born", "condemn", "pass away", "parade", "arrest jail", "carry", "vote", "start organization", "encounter", "hurt", "communication", "merge organization", "go out of business", "attend the trial", "fine", "convict", "end organization", "sue", "separate", "acquit", "appeal", "perform", "wed", "extradite", "pardon" | Separate[breakdowns] Wed[marriages] |

Table 11: Adaptive evolution of analogous schema expansion on ACE2005 dataset.

| |
|---|
| *GPT-3.5 Input Example:* |
| There are some relation extraction samples, relation must be taken from schema, head entity and tail entity must be taken from context. Relation, head entity and tail entity may have multiple. schema: ["people", "country", "religion", "major shareholder of", "industry", "contains", "brith place", "location", "nationality", "advisors", "neighborhood of", "place lived", "capital", "geographic distribution", "teams", "major shareholders", "place of death", "children", "company", "profession", "place founded", "founders"] Context: In Queens, North Shore Towers, near the Nassau border, supplanted a golf course, and housing replaced a gravel quarry in Douglaston. The relation involved in the above sentence are: 1. The head entity is Douglaston, relation is neighborhood of, tail entity is Queens; 2. The head entity is Queens, relation is contains, tail entity is Douglaston. Context: Martin, the district attorney for Lehigh County in Pennsylvania, said that after his office's review of the records, he was satisfied with Mr. Cullen's denials. The relation involved in the above sentence are: 1. The head entity is Pennsylvania, relation is contains, tail entity is Lehigh County. Context: Mr.Brown has demeaned Mr.Bush as "a cheerleader," declared that Homeland Security Secretary Michael Chertoff did not know "the first thing about running a disaster," and called critics like Representative Gene Taylor, Democrat of Mississippi, "a little twerp" and Senator Norm Coleman, Republican of Minnesota, an unprintable vulgarity (both in Playboy). The relation involved in the above sentence are: 1. The head entity is Gene Taylor, relation is place lived, tail entity is Mississippi. ... Do you understand how to do relation extraction based on schema? Now it's your turn to do relation extraction. schema: ["people", "country", "religion", "major shareholder of", "industry", "contains", "birth place", "location", "nationality", "advisors", "neighborhood of", "place lived", "capital", "geographic distribution", "teams", "major shareholders", "place of death", "children", "company", "profession", "place founded", "founders"] Context: But that spasm of irritation by a master intimidator was minor compared with what Bobby Fischer, the erratic former world chess champion, dished out in March at a news conference in Reykjavik, Iceland. The relation involved in the above sentence are: |
| *GPT-3.5 Output Example:* The relation involved in the above sentence are: 1. The head entity is Bobby Fischer, relation is place lived, tail entity is Iceland; 2. The head entity is Iceland, relation is contains, tail entity is Reykjavik; 3. The head entity is Iceland, relation is capital, tail entity is Reykjavik. |
| *Golden Output Example:* The relation involved in the above sentence are: 1. The head entity is Bobby Fischer, relation is nationality, tail entity is Iceland; 2. The head entity is Iceland, relation is capital, tail entity is Reykjavik; 3. The head entity is Iceland, relation is contains, tail entity is Reykjavik; 4. The head entity is Bobby Fischer, relation is place of death, tail entity is Reykjavik. |

Table 12: Examples of GPT-3.5 experiment on NYT dataset. A total of 20 demonstrations are given to the model.

There are some relation extraction samples, relation must be taken from schema, head entity and tail entity must be taken from Context. relation, head entity and tail entity may have multiple.

schema: ["people", "country", "religion", "major shareholder of", "industry", "contains", "brith place", "location", "nationality", "advisors", "neighborhood of", "place lived", "capital", "geographic distribution", "teams", "major shareholders", "place of death", "children", "company"]

Context: In Queens , North Shore Towers , near the Nassau border , supplanted a golf course , and housing replaced a gravel quarry in Douglaston .
The relation involved in the above sentence are: 1.The head entity is Douglaston, relation is neighborhood of, tail entity is Queens;2.The head entity is Queens, relation is contains, tail entity is Douglaston;

Context: In his authoritative and tough-minded new book , " The Assassins ' Gate : America in Iraq , " the New Yorker writer George Packer reminds us that the decision of the Bush administration to go to war against Iraq and its increasingly embattled handling of the occupation were both predicated upon large , abstract ideas about the role of America in the post-cold war world -- most notably , a belief in pre-emptive and unilateral action , the viability of exporting democracy abroad , the urge to streamline the military and the dream of remaking the Middle East .
The relation involved in the above sentence are: 1.The head entity is Middle East, relation is contains, tail entity is Iraq;2.The head entity is Middle East, relation is contains, tail entity is Iraq;

Context: Martin , the district attorney for Lehigh County in Pennsylvania , said that after his office 's review of the records , he was satisfied with Mr. Cullen 's denials .
The relation involved in the above sentence are: 1.The head entity is Pennsylvania, relation is contains, tail entity is Lehigh County;

Context: Mr. Brown has demeaned Mr. Bush as " a cheerleader , " declared that Homeland Security Secretary Michael Chertoff did not know " the first thing about running a disaster , " and called critics like Representative Gene Taylor , Democrat of Mississippi , " a little twerp " and Senator Norm Coleman , Republican of Minnesota , an unprintable vulgarity ( both in Playboy ) .
The relation involved in the above sentence are: 1.The head entity is Gene Taylor, relation is place lived, tail entity is Mississippi;

Context: Pakistan 's president , Pervez Musharraf , and President Hamid Karzai talked by telephone for an hour on Tuesday , a day after the Afghan intelligence service said it had arrested three Pakistanis who were trying to assassinate the American ambassador , Zalmay Khalilzad . "
The relation involved in the above sentence are: 1.The head entity is Pervez Musharraf, relation is nationality, tail entity is Pakistan;

Figure 7: ChatGPT input example on NYT dataset. A total of 20 demonstrations are given to the model.

HU Do you understand how to do relation extraction based on schema?
Now it's your turn to do relation extraction.
schema: ["people", "country", "religion", "major shareholder of", "industry", "contains", "brith place", "location", "nationality", "advisors", "neighborhood of", "place lived", "capital", "geographic distribution", "teams", "major shareholders", "place of death", "children", "company"]
Context: But that spasm of irritation by a master intimidator was minor compared with what Bobby Fischer , the erratic former world chess champion , dished out in March at a news conference in Reykjavik , Iceland .
The relation involved in the above sentence are: 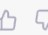

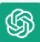

1. The head entity is Bobby Fischer, relation is people, tail entity is former world chess champion
2. The head entity is Reykjavik, relation is location, tail entity is Iceland.

HU Do you understand how to do relation extraction based on schema?
Now it's your turn to do relation extraction.
schema: ["people", "country", "religion", "major shareholder of", "industry", "contains", "brith place", "location", "nationality", "advisors", "neighborhood of", "place lived", "capital", "geographic distribution", "teams", "major shareholders", "place of death", "children", "company", "profession", "place founded", "founders"]
Context: But that spasm of irritation by a master intimidator was minor compared with what Bobby Fischer , the erratic former world chess champion , dished out in March at a news conference in Reykjavik , Iceland .
The relation involved in the above sentence are:

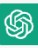

1. The head entity is Bobby Fischer, relation is people, tail entity is former world chess champion.
2. The head entity is Bobby Fischer, relation is profession, tail entity is chess player.
3. The head entity is Reykjavik, relation is location, tail entity is Iceland.
4. The head entity is Bobby Fischer, relation is place lived, tail entity is Iceland (assuming Bobby Fischer lived in Iceland at the time of the news conference)
5. The head entity is Bobby Fischer, relation is place founded, tail entity is United States (assuming Bobby Fischer was born in the United States)

Figure 8: ChatGPT output results under a horizontal schema expansion iteration.