# OpenReview forum: "Schema-adaptable Knowledge Graph Construction"
_EMNLP/2023/Conference — EMNLP 2023 Findings_

### Official Review · Reviewer_n6r2 · 2023-08-04

**Soundness:** 3

**Excitement:**

3: Ambivalent: It has merits (e.g., it reports state-of-the-art results, the idea is nice), but there are key weaknesses (e.g., it describes incremental work), and it can significantly benefit from another round of revision. However, I won't object to accepting it if my co-reviewers champion it.

**Paper Topic And Main Contributions:**

SUMMARY: The paper defines the task of "schema-adaptable knowledge graph (KG) construction", whose aim is to extract <typed head entity, relation, typed tail entity> KG triplets from natural language text when the target schema (ontology TBox) changes in time with the addition of new classes/properties lacking training instances, and without re-training/tuning of the extraction model. A methodology to derive datasets for this task is provided along with datasets derived from FEW-NERD, NYT, ACE2005. A dedicated 'baseline' model for the task, named ADAptive Knowledge Graph Construction (ADAKGC) and based on pre-trained T5 language model fed with <continuous soft prompt, current schema encoded, input text>, is proposed and evaluated over such datasets against other state-of-the-art approaches (TANL, TEXT2EVENT, UIE) that might be used for the task, showing marginal improvements over UIE in most settings.

CONTRIBUTIONS:

* C1. Definition of "schema-adaptable KG construction" task along with related concepts (e.g., horizontal/vertical/hybrid schema expansion), dataset derivation and evaluation methodologies;

* C2. Provision of a concrete evaluation benchmark for the new task derived from FEW-NERD, NYT, ACE2005, coming with data and/or code where data cannot be shared (e.g., ACE2005);

* C3. Proposal of the ADAKGC extraction model tailored to the new task for which it sets strong baseline performance, coming along with code ensuring reproducibility.

POST-REBUTTAL UPDATE. I thank the authors for the clarifications in their rebuttal. After reading it and considering the other reviews as well, I confirm the original scores in my review.

**Questions For The Authors:**

* A. How large a schema can be reasonably processed via ADAKGC, considering it has to be supplied in input to the LM? Is this a limitation to discuss in the corresponding section of the paper?

* B. For vertical expansion, when selecting a super-class C1 but not one of its sub-classes C2, are train/dev/test examples for C2 (e.g., mentions of C2 in the text, relations involving mentions of C2, etc) included among corresponding train/dev/test examples to the super-class C1? I would expect so, in order to guarantee completeness of text annotations w.r.t. class C1 (the rationale is that an occurrence in text of a mention/relation/etc for class C2 is *also* an occurrence for class C1).

**Reasons To Accept:**

* S1. Interesting and adequately motivated new task that, along with the supplied framework / benchmark / baseline, may spark interest in further work in this direction

* S2. Good reproducibility, with involved code/data resources made available to reviewers as supplemental material

* S3. Sufficient experimental validation, considering multiple KG construction sub-tasks and settings (Tab. 1,2,3) and including ablation tests (Fig 5), error analysis (Tab. 4), check for catastrophic forgetting (Fig. 6) and additional tests of GPT-3.5 / ChatGPT and of a different "analogous" schema expansion form in the Appendix.

**Reasons To Reject:**

* W1. Improvements of ADAKGC over UIE are mostly marginal, this happening also after several schema evolution iterations, i.e., the setting for which ADAKGC is specifically designed for (differently from UIE); besides this, the currently assessed performance loss incurred with schema evolution appear substantial, raising some concern about the practical feasibility of the task.

* W2. Except for schema-conditioned decoding, the architecture of ADAKGC appears to me close to the one of UIE and thus not particularly novel; besides, I wonder about the feasibility of encoding the whole schema in the input of the language model, which seems to me may pose a limit to the size of the schema that can be used in practice (see question A).

* W3. Definitions and examples of horizontal and vertical expansions appear a bit unclear. Can sub-relations be introduced as well for vertical expansion? The paper mentions and appear to focus on sub-classes, but the displayed example (Fig. 2) is actually about a sub-relation ("capital", refining "contains") and this kind of evolution would seem natural to me as well. Similarly, for horizontal expansion, what does it mean exactly for the new class to be "at the same level"? It cannot have a super-class as otherwise it would be a vertical expansion, so I imagine it has to be a new top-level class. And when adding it, can it be related to existing classes through some existing relation? All these details, along with what to do with train/test/dev examples of super-classes(/relations?) when doing vertical expansions, should be rigorously defined in the paper in order to delimit the kind of expansions considered and tested in this work.

**Reproducibility:**

5: Could easily reproduce the results.

**Reviewer Confidence:**

2: Willing to defend my evaluation, but it is fairly likely that I missed some details, didn't understand some central points, or can't be sure about the novelty of the work.

**Typos Grammar Style And Presentation Improvements:**

Schema expansion in §2.3 and Fig. 2 should be better introduced: in Fig. 2, "(people, profession, person)" looks weird (replace class "person" with class "job"?) and it's also unclear the difference between classes "people" and "person"; in §2.3 the addition of sub-class nodes is mentioned (e.g., "father") but Fig. 2 shows the addition of sub-relations (e.g., "capital").

* T1.  [§1] "adapt dynamic schema" -> "adapt to"
* T2.  [§2.1, references] check entry "(Expressions, 2005)" (wrong author)
* T3.  [§2.2] "training paramerer on" -> "trained on"
* T4.  [§2.3] "without new classes as training data" -> "instances"
* T5.  [§2.3] "and except for" -> "besides"
* T6.  [Alg. 1] please introduce the meaning of "raw" (whole original dataset?), of "W_c" and "W_s" (clear only after checking the appendix)
* T7.  [§3.1] "object" -> "objective"
* T8.  [Eq. 2] what is "Y_index"?
* T9.  [§3.3] "token by the token" -> remove "the"
* T10. [§4.4] "bellows" -> "below"
* T11. [Tab. 4] "Meet[visited]" appears to refer to mention "visited" that do not appear in the corresponding "Input Example"
* T12. [Limitations] "the proposed", "suffer", "schema-datable" -> "the proposed one", "suffers", "schema-adaptable"
* T13. [Alg. 2,3,4 in appendix] seems to me that selection of "n_init" nodes cannot simply be "random", if a proper, connected hierarchy of classes has to be extracted to start with

---

> ### Author Rebuttal · Authors · 2023-08-29
>
> We thank you for vour comments and feedback, In addition to the general updates. we address vour concerns here:
>
> > Question A: How large a schema can be reasonably processed via ADAKGC, considering it has to be supplied in input to the LM? Is this a limitation to discuss in the corresponding section of the paper?
>
> The size of the schema that can be reasonably processed via ADAKGC depends on the maximum input length that LLMs, such as GPT-3.5, can handle.  GPT-3.5 has a maximum token limit, and both the schema-enriched prefix instructor and schema-conditioned dynamic decoding need to fit within this limit along with the rest of the input text.   In the corresponding section of the paper, we will acknowledge this limitation and discuss it as a consideration for the practical application of ADAKGC.  We will mention that the maximum token limit of the LM imposes a constraint on the complexity and size of the schema that can be utilized effectively.  Additionally, we will discuss potential strategies for handling larger schemas, such as prioritizing key schema elements or implementing techniques for schema summarization.   It's important to highlight that while there is a limitation on the schema size, ADAKGC's effectiveness in handling evolving schemas is still significant and relevant for scenarios where the schema is dynamically changing or evolving over time.
>
> > Question B: For vertical expansion, when selecting a super-class C1 but not one of its sub-classes C2, are train/dev/test examples for C2 (e.g., mentions of C2 in the text, relations involving mentions of C2, etc) included among corresponding train/dev/test examples to the super-class C1? I would expect so, in order to guarantee completeness of text annotations w.r.t. class C1 (the rationale is that an occurrence in text of a mention/relation/etc for class C2 is also an occurrence for class C1).
>
> You are correct in your expectation.  In the vertical schema expansion, when selecting a super-class C1, train/dev/test examples for its sub-classes (like C2) are indeed included among the corresponding train/dev/test examples for the super-class C1. Since an occurrence in the text of a mention, relation, or other relevant information for class C2 can also be considered as an occurrence for class C1, including these examples in the context of C1 helps capture the broader context and potential relationships between different classes in the schema hierarchy.
>
> > Typos Grammar Style And Presentation Improvements
>
> We appreciate your attention to detail about typos, grammar, style, and presentation of the paper. We will thoroughly review the paper again to address the issues you've highlighted and ensure that it meets the standards of clarity and professionalism.

---

### Official Review · Reviewer_rg33 · 2023-08-06

**Soundness:** 3

**Excitement:**

3: Ambivalent: It has merits (e.g., it reports state-of-the-art results, the idea is nice), but there are key weaknesses (e.g., it describes incremental work), and it can significantly benefit from another round of revision. However, I won't object to accepting it if my co-reviewers champion it.

**Paper Topic And Main Contributions:**

This paper proposes a new task named schema-adaptable knowledge graph construction, whose goal is to perform information extraction with an evolving schema without re-training the model. The authors design three types of schema evolution (horizontal, vertical, and hybrid) and build task-specific benchmarks using existing datasets. They also propose a new model, named AdaKGC, which includes two new components: schema-enriched prefix instructors and a schema-conditioned dynamic decoding strategy, addressing the challenges of schema evolution. The empirical results on three IE tasks (NER, RE, and EE) across three datasets demonstrate the effectiveness of the proposed model.

**Questions For The Authors:**

1. If I understand the proposed task "schema-adaptable KGC" correctly, the model is only trained once with the initial schema and then evaluated on the evolving schema over different iterations. In lines 289-298, the authors introduce a three-step training strategy. May the authors confirm that this strategy is only used for the initial training? Also, the authors do not explain the rationale behind this strategy nor ablate its effect. It would be better if the authors can provide some analysis of this strategy.
2. In lines 323-325, the authors mention that "we constrain the model to generate the type tokens consistently with the existing schema at the type location." I am wondering how this is implemented. Is this something as simple as constraining the decoding vocabulary? If yes, it would be better to explicitly state it in the paper, and it is a common practice and not a novel contribution.

**Reasons To Accept:**

1. The paper studies an interesting and important task, which has great practical value.
2. The authors conduct extensive experiments and provide necessary analyses to demonstrate the effectiveness of the proposed model.

**Reasons To Reject:**

1. The writing of the paper can be improved, for example, the description of the experimental setting and model details is not clear enough, which leads to some confusion (see the questions below).
2. The technical novelty of the paper is limited as the two proposed components seem to be very straightforward.

**Reproducibility:**

4: Could mostly reproduce the results, but there may be some variation because of sample variance or minor variations in their interpretation of the protocol or method.

**Reviewer Confidence:**

4: Quite sure. I tried to check the important points carefully. It's unlikely, though conceivable, that I missed something that should affect my ratings.

---

> ### Author Rebuttal · Authors · 2023-08-29
>
> We thank you for vour comments and feedback, In addition to the general updates. we address vour concerns here:
>
> > Q1: If I understand the proposed task "schema-adaptable KGC" correctly, the model is only trained once with the initial schema and then evaluated on the evolving schema over different iterations. In lines 289-298, the authors introduce a three-step training strategy. May the authors confirm that this strategy is only used for the initial training? Also, the authors do not explain the rationale behind this strategy nor ablate its effect. It would be better if the authors can provide some analysis of this strategy.
>
> Yes, that's correct.  In the proposed task of "schema-adaptable KGC", the model is trained initially with a initial schema.  However, the novelty lies in its ability to adapt to dynamically changing schemas during evaluation without requiring re-training.  This means that the model is evaluated on new or evolving schemas in different iterations without undergoing a full training process again.  The focus is on the model's capability to extract entity, relation, and event information from the changing schema over time.
>
> The three-step training strategy introduced in lines 289-298 is indeed applied only during the initial training phase.  This strategy is designed to facilitate the convergence of our model's parameters by progressively incorporating different types of objectives( Prefix instructor, Schema-specific instructor, LM parameters). This strategy comes from our intuition that along the stepwise task learning or curriculum learning, task prefix, schema prefix, and full parameter training need to have a reasonable training sequence.
>
> > Q2: In lines 323-325, the authors mention that "we constrain the model to generate the type tokens consistently with the existing schema at the type location." I am wondering how this is implemented. Is this something as simple as constraining the decoding vocabulary? If yes, it would be better to explicitly state it in the paper, and it is a common practice and not a novel contribution.
>
> Ensuring the model can generate type tokens consistently with the existing schema involves constraining the decoding process. Thus, constraining the decoding vocabulary is an effective method where we construct vocabulary based on the dynamically changing Tire-Tree. It is indeed simple in terms of innovation and we will state it in the paper.

---

### Official Review · Reviewer_hsKx · 2023-08-12

**Soundness:** 4

**Excitement:**

4: Strong: This paper deepens the understanding of some phenomenon or lowers the barriers to an existing research direction.

**Paper Topic And Main Contributions:**

The paper addresses the challenge of schema-adaptable knowledge graphc completion (which is really knowledge extraction in this context). The fundamental premise of the paper is that the ontology of a domain might change over time. This entails that being able to adapt to changes in the ontology is necessary for the application of knowledge extraction in the real world. The authors presents novel datasets to evaluate the performance of existing extraction approaches and present a novel approach dubbed AdaKGC. In their evaluation setting, the authors refrain from retraining the underlying models. The authors compare their approach with UIE, TANL and TEXT2EVENT. The results suggest that their approach perform at least as well as the state of the art.

**Questions For The Authors:**

1) What is $S$ exactly?
2) Are your results statistically significant?
3) What are the runtimes of your approaches, especially compared to that of the reference approaches?
4) Why do you only compare with those three baselines?

**Reasons To Accept:**

The problem is highly relevant to real-life applications. The paper is easy to understand (up to some details in the equations) even if it does contain a number of typos. The evaluation is also clear.

**Reasons To Reject:**

The authors use the schema $S$ in equations without formally defining what that is. Note that the definition of schema uses a cursive S and small letters for instances. It is really unclear how that is to be understood but I guess it's an easy fix. There is no statistical test, hence it is unclear whether the performance of AdaKGC is really superior to that of the state of the art. I would assume that it is the case but would really like to see a corresponding test carried out.

**Reproducibility:**

4: Could mostly reproduce the results, but there may be some variation because of sample variance or minor variations in their interpretation of the protocol or method.

**Reviewer Confidence:**

3: Pretty sure, but there's a chance I missed something. Although I have a good feel for this area in general, I did not carefully check the paper's details, e.g., the math, experimental design, or novelty.

---

> ### Author Rebuttal · Authors · 2023-08-29
>
> We thank you for vour comments and feedback, In addition to the general updates. we address vour concerns here:
> > Q1: What is $S$ exactly?
>
> In the context of the paper, the term "schema" represented by the $S$ typically refers to the structure and organization of the knowledge graph, including the types of entities, relations, and events that are part of the graph. It encompasses the underlying framework that defines how different pieces of information are connected in the graph. Specifically, we adopt the same definition as UIE, which utilizes a specific pre-defined schema as the structural schema instructor. An innovation is that the schema we propose is dynamically constructed, demonstrated on Table(8-11) with 4 types of expansion patterns. We will provide a formal definition or explanation for the schema denoted by  $S$. We are sorry for the ambiguity of the expression and will correct these defects.
>
> > Q2: Are your results statistically significant?
>
> For each iteration experiment, we report the average performance over 3 random seeds. We conduct a pair-wise t-test and the p-values are less than 0.005
>
> > Q3: What are the runtimes of your approaches, especially compared to that of the reference approaches?
>
> The paper does not explicitly mention the specific runtimes of the approaches mentioned (Text2Event, TANL, UIE, GPT-3.5, and ADAKGC) or provide a direct comparison of their runtimes. Since real-time capability is not the highlight that we want to highlight in our paper. Intuitively, AdaKGC's runtime does not differ significantly from those baseline.
>
> > Q4: Why do you only compare with those three baselines?
>
> Text2Event, TANL, UIE, and GPT-3.5 are chosen due to their relevance to knowledge graph construction tasks. Additionally, these methods could have been widely discussed and benchmarked in previous literature, making them suitable for evaluation against the proposed schema-adaptable KGC approach, AdaKGC. A detailed description of them is shown in Section 2.4 Vanilla Baselines (pages 3-4).

---

### Meta-Review · Area_Chair_ji8n · 2023-09-19

**Recommendation:** 4

**Metareview:**

The paper proposes a new task named schema-adaptable knowledge graph construction to enable the model to handle emerging knowledge without retraining the IE models. This work is applicable in real-world scenarios. The authors have studied changes in the schema in three dimensions: horizontal, vertical, and hybrid, and they have built task-specific benchmarks using existing datasets.

The proposed model is evaluated on three IE tasks (NER, RE, and EE) across three datasets and compared to four baselines, achieving similar scores as the baselines. However, the t-test indicates that the results are statistically significant with a p-value < 5%.

Paper is easy to follow with typos and grammar style errors.

---

### Decision · Program_Chairs · 2023-10-07

**Decision:**

Accept-Findings

**Comment:**

The paper proposes a new task named schema-adaptable knowledge graph construction to enable the model to handle emerging knowledge without retraining the IE models. This work is applicable in real-world scenarios. The authors have studied changes in the schema in three dimensions: horizontal, vertical, and hybrid, and they have built task-specific benchmarks using existing datasets.

The proposed model is evaluated on three IE tasks (NER, RE, and EE) across three datasets and compared to four baselines, achieving similar scores as the baselines. However, the t-test indicates that the results are statistically significant with a p-value < 5%.

Paper is easy to follow with typos and grammar style errors.